# Heavy Metal Contamination and Potential Ecological Risk Assessment in Sediments of Lake Bafa (Turkey)

**Aykut Yozukmaz \*** and **Murat Yabanlı**

Department of Aquatic Sciences, Faculty of Fisheries, Mugla Sitki Kocman University, 48000 Mugla, Turkey; muratyabanli@mu.edu.tr
\* Correspondence: aykutyozukmaz@mu.edu.tr

**Abstract:** This study examined the spatio-temporality of heavy metal concentrations (Al, Cd, Co, Cr, Fe, Mn, Ni, Pb and Zn) in the sediments of Lake Bafa, one of the most important wetlands of Turkey's Aegean region. The study evaluated sediment quality according to threshold effect concentration (TEC) and probable effect concentration (PEC) values based on sediment quality guidelines (SQG), and provided a potential ecological risk assessment (PERI) along with indices such as geoaccumulation index ($NI_{geo}$), enrichment factor (*EF*), contamination factor (*CF*), and pollution load index (PLI). For this purpose, surface sediment from 10 different points and core samples from three different points were seasonally collected and the concentrations of nine heavy metals were determined by ICP-MS. The findings indicated the following accumulation order of heavy metals in the sediment: Fe > Al > Mn > Ni > Cr > Zn > Pb > Co > Cd, with concentrations of Al, Mn, and Ni being high in the surface sediment samples. According to the $NI_{geo}$, surface sediment and core samples were very slightly polluted with Cr, Mn, and Co at most stations, while five stations were slightly polluted with Cd. Regarding EF, the lake was at risk in terms of Al and Pb accumulation. The CF results indicated that the lake was under pressure in terms of heavy metal pollution. The PLI results indicated a significant pollution hazard at all stations, while the PERI analysis indicated moderate risk of heavy metal pollution at some stations. As one of the most comprehensive studies applying such indices to Lake Bafa, the results are very significant in terms of evaluating the lake's ecological sustainability.

**Keywords:** Lake Bafa; geoaccumulation index; enrichment factor; contamination factor; pollution load index; potential ecological risk assessment

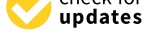



## 1. Introduction

During the 20th century, rapid population-driven socio-economic development led to an increase in urbanization, industrial development, and agricultural activities, and a concomitant increase in demand for clean water resources [1–3]. The industrial revolution caused a significant increase in pollution in aquatic ecosystems to become the most important current environmental issue [4–9].

The pollutants that adversely affect natural ecosystems enter the aquatic ecosystem from various sources and are exposed to physical, chemical, and biological processes [10]. These processes directly depend on the structure of metals in their environment, their concentrations, and the metabolic activities of living organisms exposed to this pollution. Other physical factors, such as suspended solids (SS), temperature, dissolved oxygen, and pH, also affect the circulation properties of pollutants [11]. The most dangerous pollutants for the natural environment are those that remain intact for a long time in their environment, cannot be assimilated, and have highly toxic effects [12,13].

Such pollutants include heavy metals, which can be toxic to organisms at high concentrations [14], do not decompose, and infuse up the food chain into higher level

organisms to accumulate in various tissues and organs in a process called bioaccumulation. The ultimate accumulation levels in equivalent tissues and organs in different organisms can vary depending on their structures, leading to different alterations in particular tissues and organs [15]. Once concentrations of heavy metals in an ecosystem increase beyond trace amounts, living organisms cannot metabolize them efficiently. Therefore, their bioaccumulation increases faster than their environmental concentrations, leading to the transportation of heavy metals to higher levels in the food chain [4,16,17]. As a result, when heavy metals exceed their natural concentrations in aquatic ecosystems, they have many different adverse effects that limit the vital functions of living organisms.

Having entered the aquatic ecosystem in many different ways, heavy metals infuse into the food chain through various environmental components (water, sediment, seston, etc.) and living organisms (micro and/or macro vertebrates and invertebrates), or through non-nutritive ways (e.g., respiration, absorption through the skin, adsorption) [18,19]. Although these processes do not directly damage living organisms, heavy metals accumulate in different tissues and organs due to bioaccumulation and complex food chain interactions [20,21]. This bioaccumulation occurs because the rate of metabolic removal of pollutants directly or indirectly taken in by living organisms in aquatic ecosystems is slower than their uptake rate. The term bio-concentration refers to the level of substances that living organisms take directly from the water through different tissues and organs (gills, epithelial tissue, etc.) which then accumulate in tissues and organs (muscle, kidney, liver, etc.) [22–24]. Due to bioaccumulation, living organisms that perform their vital functions in aquatic ecosystems can accumulate pollutants at much higher amounts than the pollutants' concentrations in the water itself [25].

After entering the aquatic ecosystem, heavy metals do not remain in the water column for long if their concentrations are high. Instead, they settle into the sediment [26–28]. While heavy metals adsorbed in sediment are not a direct source for aquatic organisms, they can be released back into the water column due to environmental changes (e.g., in temperature, salinity, pH, redox potential) occurring in the constantly dynamic water column above the sediment [29]. Consequently, sediment acts as a renewable resource for aquatic ecosystems for such pollutants [30]. Furthermore, due to their structure, both organic and inorganic pollutants can endure aquatic environmental conditions and are not decomposed by physical, chemical, and biological processes. Therefore, they can accumulate in the sediment layer over many years to pose both a direct and indirect threat to the health of humans and aquatic organisms. The sediment layers of aquatic ecosystems affected by urbanization contain particularly high levels of pollutants [31,32] that lead to severe environmental problems [33,34]. Therefore, it is important to protect sediment quality to ensure the sustainability of aquatic life and ecological balance, and to biologically protect water bodies that cross national borders, whether small or large in volume. Pollutants in the sediment can threaten or even eliminate aquatic species by damaging the food chain. Due to physical, chemical, and biological processes occurring in the sediment, pollutant groups can also be transferred into the water layer and living organisms due to bioaccumulation along the food chain [35].

Thus, given that the quality of the sediment layers helps determine the quality of the water column above it [36,37], environmental sedimentology studies and water quality analyses should be carried out simultaneously. More specifically, all pollutants (organic substances, phosphates, nitrogenous compounds, and various metals) have specific saturation levels in the sediment layer. Once their sedimentary concentrations reach this level, they are released into the water as a pollutant source. Thus, it is not enough to focus on solving pollution limited to the water column because pollution can reoccur due to the release of pollutants from the sediment [38]. Chemical analysis can be used to determine the sediment layer's environmental risk level and establish standard quality criteria for sediment quality. The pollutant concentrations can be compared to the toxicity levels in living organisms in the sediment and the substances that affect them [39]. In short, regular examination of sediment is important in determining the level of risk in aquatic ecosystems.

The present study examined the spatio-temporal concentrations of nine heavy metals (Al, Cd, Co, Cr, Fe, Mn, Ni, Pb, and Zn) in the sediments of Lake Bafa, which is one of the most important wetlands of Turkey's Aegean region. Sediment quality was evaluated according to threshold effect concentration (TEC) and probable effect concentration (PEC) values based on sediment quality guidelines (SQG), and with a potential ecological risk assessment (PERI), along with the following four indices: geoaccumulation index ($NI_{geo}$), enrichment factor (*EF*), contamination factor (*CF*), and pollution load index (PLI). Lake Bafa was selected as the sampling area because it has a unique aquatic ecosystem connected both to the sea and the groundwater system, and is affected by several anthropogenic factors, particularly agricultural, domestic, and industrial waste. In addition, with the current study, it will be possible to have an idea about how the accumulation of toxic pollutants in aquatic ecosystems with similar ecological characteristics in the world will affect the quality of lake sediment.

## 2. Material and Method

### 2.1. Area of Study

The sampling area of this study was Lake Bafa, located in Turkey's Aegean region between the provinces of Aydın and Muğla ($37°30'$ N, $27°25'$ E). The lake's water surface covers approximately 6708 ha. The lake's north-south width is 4.5 km, while its east-west length is 15.4 km. The lake surface is 10 m above sea level, while its deepest point is 21 m. The lake's most important freshwater sources are surface and underground waters from the Büyük Menderes River and the Beşparmak Mountains around the lake's north-east shore [40].

Lake Bafa was once a bay connected to the Aegean Sea. Marine conditions were dominant in the lake until the Hellenistic period [41], when the seaward connection was lost due to alluvial transport in the Büyük Menderes River, which formed a natural barrier (coastal dam) lake ecosystem over a period of almost 6000 years. The region where Lake Bafa is located has always been an important center for civilizations throughout history as both a marine ecosystem and then a freshwater ecosystem. Nowadays, the lake is a tourist attraction due to its historical features drawn from many different civilizations and its biological diversity.

In 1985, an earthen embankment was built by the General Directorate of State Hydraulic Works at the point where the Büyük Menderes River enters the north-west side of the lake. Given that it is the largest river in Western Anatolia with a total length of 584 km that flows through Afyon, Denizli, Uşak, and Aydın provinces before discharging into the Aegean Sea [42], the embankment separated Bafa Lake from its main fresh water source and caused irreversible changes to its ecosystem. In 1994, the lake and surrounding forestland were designated as a national park named "Menderes Delta National Park". In addition, the areas within the park containing archaeological artifacts were designated as Grade 1 Protected Areas [43]. In recognition of its rich biodiversity, Lake Bafa is also listed as an Area of Special Interest under the Ramsar and Bern international conventions.

Cutting the inflow of fresh water from the Büyük Menderes River led to an increase in salinity levels that has gradually transformed all the lake's fauna and flora. In addition, the large human population (approximately 2.5 million people live alongside the Büyük Menderes River) and surrounding small-scale industrial activities (especially olive oil factories) have led to the discharge of both domestic and industrial waste into Lake Bafa [44].

### 2.2. Sample Collection

The sampling strategy was designed to include bottom sediments of all major river courses by referring to similar studies conducted in the region. Accordingly, sediment samples were seasonally collected between December 2013 and November 2014 from 10 predetermined stations (Figure 1 and Table 1).

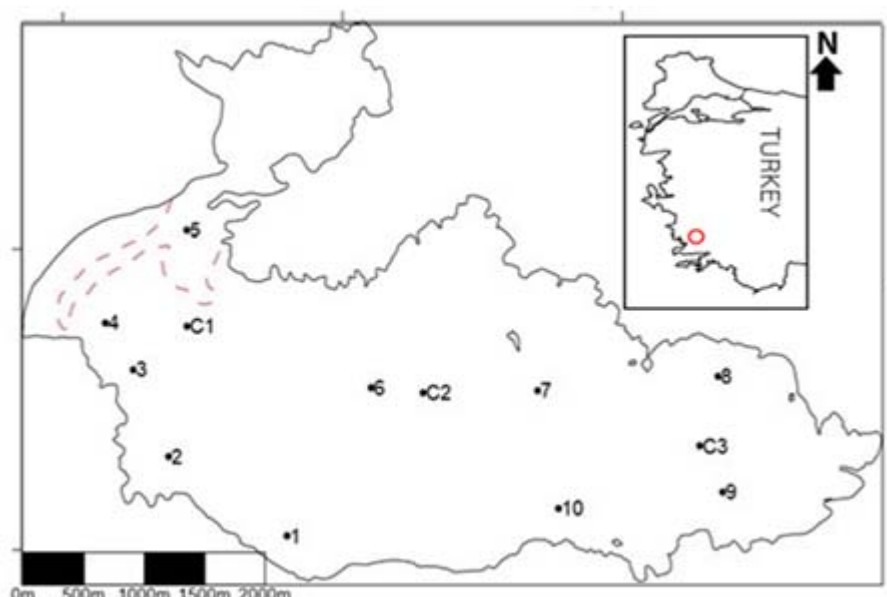

**Figure 1.** Sediment and core (C) sampling station locations in Lake Bafa basin.

**Table 1.** Sediment and core (C) sampling station locations and characteristics in Lake Bafa.

| Station | Coordinates | Features/Depth (m) |
|---|---|---|
| Station 1 | 37°28′4″ N–27°27′37″ E | Restaurant Location—4 |
| Station 2 | 37°29′36″ N–27°24′59″ E | Near Highway—21 |
| Station 3 | 37°30′01″ N–27°23′56″ E | Olive Tree Field 1—11 |
| Station 4 | 37°30′30″ N–27°23′13″ E | Lake Bafa Stream Mouth and Fish Farm Location—8 |
| Station 5 | 37°31′19″ N–27°24′08″ E | Büyük Menderes Mouth and Serçin Village Location—3 |
| Station 6 | 37°30′06″ N–27°26′35″ E | Middle of the Lake—18 |
| Station 7 | 37°30′13″ N–27°27′11″ E | Menet Isle—9 |
| Station 8 | 37°30′06″ N–27°30′31″ E | Kapıkırı Village—5 |
| Station 9 | 37°28′53″ N–27°31′15″ E | Olive Tree Field 2—5 |
| Station 10 | 37°28′59″ N–27°28′27″ E | Hotel Location—7 |
| $C_1$ | 37°31′13″ N–27°24′06″ E | Büyük Menderes Stream Mouth |
| $C_2$ | 37°30′04″ N–27°26′52″ E | Middle of the Lake |
| $C_3$ | 37°29′59″ N–27°30′34″ E | Eastern part of the Lake |

The sediment samples were collected from 5–10 cm below the sediment surface using an Ekman bottom sampler ($15 \times 15 \times 20$ = approximately 225 cm$^2$) and placed in acid-cleaned glass containers. Three additional samples ($C_1$, $C_2$ and $C_3$) were taken in 1 month using a 10-cm sediment core sampler, and examined in 2 pieces of 5 cm length. This allowed depth-wise variations in accumulation levels and particle size to be calculated. All samples were placed in an ice box during transportation in accordance with the standards and stored in appropriate laboratory conditions until the date of analysis.

In addition, measurements were taken at each sampling station of the lake's main physico-chemical parameters (temperature, pH, salinity, dissolved oxygen (DO), total dissolved solids (TDS), and conductivity) using a multiprobe water quality measurement device (YSI Professional Plus). Finally, measurements were also made in the lab of the lake's water nutrient levels (nitrite nitrogen, nitrate nitrogen, ammonium nitrogen, and phosphate phosphorus) using a DR 3900 spectrometer with suitable ready-to-use kits.

### 2.3. Heavy Metal Analysis in Sediment

Sediment samples were collected from each station seasonally for 1 year between December 2013 and November 2014 and brought to the laboratory under appropriate

conditions (in a cooler with a mean temperature of 4 °C). The sediment samples, which were stored under suitable conditions (at −18 °C) until pretreatment were removed from the deep freezer and thawed. Sub-samples of 5 g from each sample were mixed in glass containers pre-cleaned with acid, then weighed and resolved to 0.5 g aqueous sediment solutions. For heavy metal analysis, they were dissolved in 3 mL of hydrochloric acid, nitric acid, and water (HCl–HNO$_3$–H$_2$O) solution (at a ratio of 3:1:2) at 95 °C for 1 h and diluted with 10 mL of distilled water. The obtained colorless solution was centrifuged at 4000 rpm for 10 min and left to cool. The supernatant was then carefully taken using a syringe and transferred to capped falcon tubes of appropriate volume. Distilled water was added to each dissolved sample to reach a final volume of 15 mL, and the samples were made ready for analysis. Measurements were made using an Agilent brand 7700× model inductive coupled mass spectrometry (ICP-MS) [45–47]. The device's detection limits for sediment are shown in Table 2.

**Table 2.** Detection limits for the Agilent 7700× ICP-MS.

| Heavy Metals | Sediment ($\mu$g kg$^{-1}$) |
|---|---|
| Al | 0.127 |
| Cd | 0.002 |
| Co | 0.002 |
| Cr | 0.036 |
| Fe | 0.125 |
| Mn | 0.037 |
| Ni | 0.805 |
| Pb | 0.121 |
| Zn | 1.483 |

The accuracy and precision of the heavy metal analysis results in the ICP-MS were checked with standard reference material (Sigma-Aldrich® CRM016- Fresh Water Sediment 3 for Trace Metals) (Table 3).

**Table 3.** Results of CRM016 Freshwater sediment certified trace metals reference material analysis (mg kg$^{-1}$).

| Element | Certified Value | Measured Value | Recovery Rate (%) |
|---|---|---|---|
| Al | 8920 ± 657 | 8110 ± 318 | 90.92 |
| Cd | 0.47 ± 0.08 | 0.43 ± 0.05 | 91.49 |
| Cr | 14.5 ± 1.36 | 15.12 ± 1.02 | 104.28 |
| Fe | 16,800 ± 517 | 15,093 ± 389 | 89.84 |
| Pb | 14.1 ± 0.66 | 13.57 ± 0.26 | 96.24 |
| Mn | 180 ± 3.65 | 166 ± 6.74 | 92.22 |
| Ni | 16.7 ± 0.50 | 14.57 ± 0.33 | 87.25 |
| Zn | 69.7 ± 2.11 | 72.88 ± 9.15 | 104.56 |
| Co | 5.96 ± 0.24 | 5.58 ± 0.37 | 93.62 |

All the data obtained were further processed with the ArcGIS Pro Desktop application and turned into a map to show the geographically significant status (see Supplementary File).

*2.4. Particle Size and Heavy Metal Analysis in Sediment Core Samples*

Particle size analyses (PSA) were performed to detect heavy metal accumulation trends in sediment core samples. For particle size measurements, 3 core samples (13 cm on average) were divided into 5 cm slices. Each slice was then divided into sub-samples of 5 g, of which 3 were dried in an oven at 105 °C for PSA. Each dried sample was then homogenized individually in a porcelain mortar and sieved serially through 5 mm, 0.3 mm, and 0.063 mm mesh sieves. This process demonstrated that there were no particles larger than 5 mm, so the PSA groups were defined as follows [48]:

(a)   >0.30 mm = coarse sand;
(b)   0.30–0.063 mm = fine grain sand;
(c)   <0.063 = clay.

The weight ratios of the three groups in the total sediment were calculated by weighing. Each group in each sample was then pretreated to make it ready for ICP-MS analysis.

### 2.5. Determination of Organic Carbon Amounts in Sediment Samples

The organic carbon amount in the sediment samples was determined with the Walkley–Black method. This is based on the combustion of all organic matter in the samples using potassium dichromate ($K_2Cr_2O_7$) and concentrated sulfuric acid at appropriate concentrations during the heat treatment of the samples, and back titration with ferro ammonium sulfate and phenylamine indicator. The carbonates and bicarbonates in the sediment samples were removed using 10% HCl. To determine the carbon content, approximately 0.2–0.5 g was taken from each sample and placed in a glass flask thoroughly cleaned with acid. Back titration was then applied with adjusted bichromate solution and iron ammonium sulfate solution [49].

### 2.6. Assessing the Contamination Status of the Sediment

The geoaccumulation index ($NI_{geo}$) is a scale developed to indicate pollution levels in coastal sediment due to anthropogenic terrestrial heavy metal accumulation [50,51]. It is calculated according to the following formula:

$$NI_{geo} = log_2 \left( C_n / 1.5 \times B_n \right) \tag{1}$$

where $B_n$ refers to the current heavy metal concentration in unpolluted sediment. $C_n$ refers to the current heavy metal concentration in the sediment sample and the coefficient (1,5) refers to the possible changes from terrestrial effects. The obtained results are categorized as follows: "$NI_{geo} < 1$, unpolluted; $1 < NI_{geo} < 2$, very slightly polluted; $2 < NI_{geo} < 3$, slightly polluted; $3 < NI_{geo} < 4$, moderately polluted; $4 < NI_{geo} < 5$, very polluted; $NI_{geo} > 5$, very much polluted". In the present study, the background values were based on the results from core sample number 2, which had the lowest concentrations for all heavy metals.

Like the geoaccumulation index, the enrichment factor (*EF*) is a scale used to determine the lithogenic effects on heavy metal concentrations in sediment [52]. It is calculated according to the following formula:

$$EF = \left( \frac{C_x}{C_{ne}} \right)_{Sample} \bigg/ \left( \frac{C_x}{C_{ne}} \right)_{Background} \tag{2}$$

where $C_x$ refers to the metal concentration calculated by the enrichment factor; $C_{ne}$ refers to the concentration of the normalizing element. When calculating *EF*, conservative elements such as Al and Fe, which are naturally found in high concentrations in the structure of the lithosphere, are used as normalizing elements [53,54]. A result of $0.5 \leq EF \leq 1.5$ indicates that heavy metal accumulation occurred due to natural processes, whereas a result of $EF > 1.5$ means that heavy metal accumulation resulted from anthropogenic processes [55]. In the present study, Fe concentrations were used as the normalizing element.

The contamination factor *(CF)* also evaluates heavy metal accumulation in sediments by comparing its level to preindustrial reference levels [56]. It is calculated according to the following formula:

$$CF = \frac{C_{(metal)}}{C_{(background)}} \tag{3}$$

where $C_{(metal)}$ refers to the concentration of the sampled metal and $C_{(background)}$ stands for the reference control value. As explained above, in the present study, heavy metal concentrations in the sediment sample taken from the 10 cm depth of the $C_2$ core sample were used as background. *CF* values are categorized into four levels of contamination: "*CF* < 1 = low con-

tamination; $1 \leq CF < 3$ = moderate contamination; $3 \leq CF < 6$ = considerable contamination; and $CF \leq 6$ = very high contamination" [54].

The pollution load index (PLI) also measures heavy metal pollution in sediments [57,58]. PLI is calculated using the following formula:

$$PLI = (CF_1 \times CF_2 \times CF_3 \times \cdots \times CF_n)^{1/n} \tag{4}$$

where CF is the contamination factor for each heavy metal under consideration, calculated according to Equation (3). PLI values are categorized as follows: "$1 > PLI$ = no contamination; $PLI = 1$ = baseline levels of contamination; and $1 < PLI$ = deterioration of site quality" [54].

### 2.7. Assessing the Potential Ecological Risk (PERI) of Sediment

Sediment quality guidelines (SQG) are used to understand whether heavy metals accumulating in the sediment pose an ecological risk. However, SQG has not previously been used to assess this risk in wetland samples (for both marine and freshwater ecosystems) in Turkey. Therefore, threshold effect concentration (TEC) and probable effect concentration (PEC) values taken from internationally accepted SQGs were used to compare the sediment quality of the samples in the present study [59–61]. TEC shows the heavy metal concentration below which negative ecological effects are not anticipated to occur [59]. Concentrations which are equal to or above the TEC but below the PEC define the range within which ecological effects rarely occur, whereas concentrations above the PEC indicate the range within which negative ecological effects are likely to occur often [60].

In addition, the potential ecological risk index (PERI) [56,61,62] is calculated to predict the potential effects of heavy metals on aquatic ecosystems. The index can reveal potential relationships (synergy, toxicity level, and ecological sensitivity) between all the assessed heavy metals [54] whose concentrations were determined in the study. PERI is calculated using the following formulae:

$$PERI = \sum_{i=1}^{n} E_r^i \tag{5}$$

$$E_r^i = T_r^i \times C_r^i \tag{6}$$

where $n$ is the number of heavy metals; $i$ is the heavy metal of interest in the sediment; $E_r^i$ is the potential ecological risk coefficient of a single heavy metal; and $T_r^i$ is the toxic response factor for the heavy metal of interest" [54,63]. $T_r^i$ values for Cd, Cr, Hg, Mn, Pb, and Zn were 30, 2, 40, 1, 5, and 1, respectively. $C_r^i$ represents the calculated $CF$ for each metal. $E_r^i$ values are interpreted as follows: "$E_r^i < 40$ = low risk; $40 \leq E_r^i \leq 80$ = moderate risk; $80 \leq E_r^i < 160$ = considerable risk; $160 \leq E_r^i < 320$ = high risk; $E_r^i \geq 320$ = very high risk" (Decena et al., 2018). PERI values are categorized as follows: "PERI < 90 = low risk; $90 \leq PERI < 180$ = moderate risk; $180 \leq PERI < 360$ = strong risk; $360 \leq PERI < 720$ = very strong risk; and PERI $\geq 720$ = very high risk" [56].

### 2.8. Statistical Analyzes

Pearson correlation tests were performed to reveal the significance of the relationship between the physico-chemical variables and the heavy metal concentrations. Analysis of variance (ANOVA) was applied to test whether there were significant differences between stations in heavy metals concentrations. Principal component analysis (PCA) was used to determine the relationship among all the environmental variables and the heavy metal concentrations. All analyses were performed with the IBM® SPSS Statistics® 24.0 program. A value of $p < 0.05$ was selected as significant.

## 3. Results

### 3.1. Analyses on Heavy Metal Concentrations in Surface Sediment and Core Samples

The results of heavy metal analysis (Al, Cr, Co, Ni, Zn, Cd, Pb, Fe, and Mn) for the sediment and core samples taken between December 2013 and November 2014 showed that the mean heavy metal concentrations in the surface sediment of Lake Bafa ranked as follows from highest to lowest: Fe > Al > Mn > Ni > Cr > Zn > Pb > Co > Cd. The detailed results for each element are presented below.

**Aluminum (Al):** The lowest aluminum concentration (54.03 mg kg$^{-1}$) was observed at Station 10 in summer, while the highest (7251. 38 mg kg$^{-1}$) was at Station 1 in summer. Al accumulation at Station 10 was significantly different from that at Stations 1, 2, 3, and 5 ($p < 0.05$). Seasonally, mean Al concentrations in the spring and summer differed from the fall and winter seasons. The core samples were segregated by sampling depth, divided into particle size subgroups using the PSA method, and analyzed for Al levels. The findings showed that Al tends to bind most to sediment particle sizes of <0.063 mm.

**Manganese (Mn):** The lowest manganese concentration (197.05 mg kg$^{-1}$) was at Station 8 in winter, while the highest (1331.45 mg kg$^{-1}$) was at Station 2 in fall. Mn levels did not differ seasonally or between sampling stations. The core samples were segregated by sampling depth, divided into particle size subgroups using the PSA method, and analyzed for Mg levels. The results showed that Mn tended to bind to sediment particles of <0.063 mm. Sediment particles of >0.3 mm at 5–10 cm depth adsorbed Mn element at high levels.

**Iron (Fe):** The lowest iron concentration (22,308.95 mg kg$^{-1}$) was detected at Station 9 in spring, while the highest (41,345.00 mg kg$^{-1}$) was detected at Station 8 in spring. Fe element accumulated in various regions and point sources. However, there were no significant differences in terms of stations and seasons. The core samples were segregated by sampling depth, divided into particle size subgroups using the PSA method, and analyzed for Fe levels. The results showed that Fe tended to bind to sediment particles of <0.063 mm in the core sample from 5–10 cm depth and adsorbed to sediment particles of 0.3–0.063 mm in the core samples from 0–5 cm.

**Chromium (Cr):** The lowest chromium concentration was LOD (below the analysis limits) at Station 10 in summer, while the highest (330.82 mg kg$^{-1}$) was at Station 7 in spring. There were no significant seasonal or spatial differences in Cr levels. Cr accumulated at certain point sources. The core samples were segregated by sampling depth, divided into particle size subgroups using the PSA method, and analyzed for Cr levels. The results showed that Cr tended to bind most to sediment particles of <0.063 mm.

**Cobalt (Co):** The lowest cobalt concentrations were LOD at Stations 1 and 6 in the fall, while the highest (0.73 mg kg$^{-1}$) were at Stations 2 and 7 in summer. There were no significant differences in Co concentrations in terms of sampling stations. There were significant seasonal differences between spring and summer, but not between fall and winter. The core samples were segregated by sampling depth, divided into particle size subgroups using the PSA method, and analyzed for Co levels. The results showed that Co tended to bind the most to sediment particles of <0.063 mm.

**Nickel (Ni):** The lowest nickel concentration (106.92 mg kg$^{-1}$) was at Station 7 in summer, while the highest was at Station 10 in summer (373.48 mg kg$^{-1}$). Ni accumulation at Station 9 was significantly different from that at Stations 1, 2, and 4, as was Ni accumulation at Station 10 from Stations 1 and 4, and Ni element at Station 8 from Station 1 ($p < 0.05$). There were no statistically significant differences in seasonal accumulations.

**Cadmium (Cd):** The lowest cadmium concentrations (0.02 mg kg$^{-1}$) were detected at Stations 5, 6, 7, and 10 in fall and at Station 1 in winter, while the highest (0.20 mg kg$^{-1}$) were at Station 10 in spring season and Station 2 in fall. Cd sedimentary accumulation did not differ significantly between stations. However, Cd accumulation was significantly higher in summer than spring and fall. The core samples were segregated by sampling depth, divided into particle size subgroups using the PSA method, and analyzed for Cd levels. The results indicated that Cd tends to bind the most to sediment particles of <0.063 mm.

**Lead (Pb):** The lowest lead concentration (11.02 mg kg$^{-1}$) was at Station 2 in summer, while the highest (27.57 mg kg$^{-1}$) was at Station 9 in spring. There were significant seasonal differences in PB sedimentary concentrations, which increased significantly in summer and winter, and decreased in spring and fall. There were also no significant differences according to sampling station. The core samples were segregated by sampling depth, divided into particle size subgroups using the PSA method, and analyzed for Pb levels. The results showed that Pb tends to bind the most to sediment particles of <0.063 mm.

**Zinc (Zn):** The lowest zinc concentration (22.03 mg kg$^{-1}$) was at Station 8 in fall, while the highest (87.17 mg kg$^{-1}$) was at Station 1 in fall. While Zn concentrations increased in winter and accumulated in different parts of the lake in other seasons, these differences were not significant for either sampling stations or seasons. The core samples were segregated by sampling depth, divided into particle size subgroups using the PSA method, and analyzed for Zn levels. The results showed that Zn tends to bind the most to sediment particles of <0.063 mm. However, for the core sample taken from 5–10 cm depth, particles of >0.3 mm adsorbed as much Zn as particles of <0.063 mm.

*3.2. Analyses of the Relationship between Total Organic Carbon (TOC) and Heavy Metal Concentrations*

TOC values were determined in surface sediment samples taken from 10 different stations from Lake Bafa (Table 4).

**Table 4.** The amount of TOC obtained from sediment samples taken from 10 Stations in Lake Bafa in four different seasons (g kg$^{-1}$). In bold are the lowest and higher values.

|  | Winter | Spring | Summer | Fall |
|---|---|---|---|---|
| Station 1 | 1.68 | 3.15 | 2.58 | 2.52 |
| Station 2 | 1.88 | 1.98 | 1.72 | 1.88 |
| Station 3 | 1.81 | 1.96 | 3.18 | 1.49 |
| Station 4 | 1.84 | 1.63 | 1.40 | 0.77 |
| Station 5 | 2.55 | 2.58 | 0.59 | 3.06 |
| Station 6 | 1.40 | 1.73 | 2.43 | 3.31 |
| Station 7 | 1.73 | 5.50 | 3.07 | 1.74 |
| Station 8 | 1.97 | 2.21 | 1.51 | 1.01 |
| Station 9 | 2.46 | 1.83 | 2.44 | 2.38 |
| Station 10 | 1.04 | 2.94 | 1.99 | 1.82 |

The results show that Station 7 in the spring provided the highest TOC input to the surface sediment, while the lowest accumulation was at Station 4 (Table 4.). Seasonal TOC accumulation can be ranked from highest to lowest as follows: Spring > Summer > Fall > Winter.

There was a significant positive correlation between mean TOC values from the surface sediment and Cr accumulated in the sediment. In addition, there were significant positive correlations between Al and Ni levels, and Cd and Co levels (Table 5).

TOC levels in Lake Bafa sediment varied between 0.59 and 5.50 g kg$^{-1}$. As a result of the correlation analyses between TOC and heavy metal concentrations in the Lake Bafa sediment, a positive correlation was determined between the amount of Cr and the amount of TOC. A strong positive correlation was also found between Al and Ni, and between Cd and Co.

*3.3. Analyses of $NI_{geo}$, EF, CF, and PLI*

$NI_{geo}$, *EF*, *CF*, and PLI were calculated for each heavy metal based on the sediment results and core samples.

For $NI_{geo}$, the results varied between <1 and 2–3 for the mean heavy metal concentrations in the surface sediment sampled from 10 different stations and the core samples from three different stations. For Al, Ni, and Mn, some stations were very slightly polluted, while all stations were very slightly polluted for Cr and Co. For Cd, some stations were slightly polluted and some stations were very slightly polluted (Table 6).

**Table 5.** Correlations between heavy metal concentrations and TOC in Lake Bafa sediment.

| | TOC | | | | | | | | | |
|---|---|---|---|---|---|---|---|---|---|---|
| TOC | 1.00 | Al | | | | | | | | |
| Al | −0.020 | 1.00 | Cr | | | | | | | |
| Cr | 0.370 * | 0.195 | 1.00 | Fe | | | | | | |
| Fe | −0.189 | −0.131 | 0.121 | 1.00 | Mn | | | | | |
| Mg | −0.069 | 0.069 | −0.020 | −0.075 | 1.00 | Ni | | | | |
| Ni | −0.221 | 0.847 ** | 0.035 | −0.212 | 0.162 | 1.00 | Pb | | | |
| Pb | 0.011 | 0.246 | 0.087 | 0.101 | 0.174 | 0.106 | 1.00 | Zn | | |
| Zn | 0.050 | −0.069 | −0.178 | 0.256 | −0.118 | −0.159 | 0.262 | 1.00 | Cd | |
| Cd | −0.091 | −0.055 | 0.006 | −0.001 | 0.146 | −0.116 | 0.125 | 0.232 | 1.00 | Co |
| Co | −0.217 | 0.213 | −0.060 | 0.239 | −0.274 | 0.033 | −0.193 | 0.103 | 0.505 ** | 1.00 |

* $p < 0.05$, ** $p < 0.01$.

**Table 6.** $NI_{geo}$ values for heavy metals detected in Lake Bafa sediment samples.

| | Al | Cr | Mn | Pb | Zn | Co | Cd | Ni |
|---|---|---|---|---|---|---|---|---|
| Station 1 | 1–2 | 1–2 | <1 | <1 | <1 | 1–2 | <1 | 1–2 |
| Station 2 | <1 | 1–2 | 1–2 | <1 | <1 | 1–2 | 1–2 | 1–2 |
| Station 3 | 1–2 | 1–2 | 1–2 | <1 | <1 | 1–2 | <1 | 1–2 |
| Station 4 | <1 | 1–2 | 1–2 | <1 | <1 | 1–2 | <1 | 1–2 |
| Station 5 | <1 | 1–2 | <1 | <1 | <1 | 1–2 | 1–2 | 1–2 |
| Station 6 | <1 | 1–2 | <1 | <1 | <1 | 1–2 | <1 | <1 |
| Station 7 | <1 | 1–2 | <1 | <1 | <1 | 1–2 | <1 | <1 |
| Station 8 | <1 | 1–2 | <1 | <1 | <1 | 1–2 | 1–2 | <1 |
| Station 9 | <1 | 1–2 | <1 | <1 | <1 | 1–2 | 1–2 | <1 |
| Station 10 | <1 | 1–2 | 1–2 | <1 | <1 | 1–2 | 1–2 | <1 |
| C1 | 1–2 | 1–2 | <1 | <1 | <1 | 1–2 | 2–3 | - |
| C2 | <1 | 1–2 | 1–2 | <1 | <1 | 1–2 | 2–3 | - |
| C3 | <1 | 1–2 | 1–2 | <1 | <1 | 1–2 | 2–3 | - |

The enrichment factors calculated from the sediment sample data varied as follows for each heavy metal: $EF_{(Al)}$ 0.4–2.0; $EF_{(Cr)}$ 0.65–1.0; $EF_{(Mn)}$ 0.62–1.38; $EF_{(Pb)}$ 1.18–1.6; $EF_{(Zn)}$ 0.8–1.18; $EF_{(Co)}$ 0.6–1.1 and $EF_{(Cd)}$ 0.37–1.13. For all stations, the $EF$ values indicated no anthropogenic enrichment ($EF < 2$) of heavy metals at most sites (Figure 2).

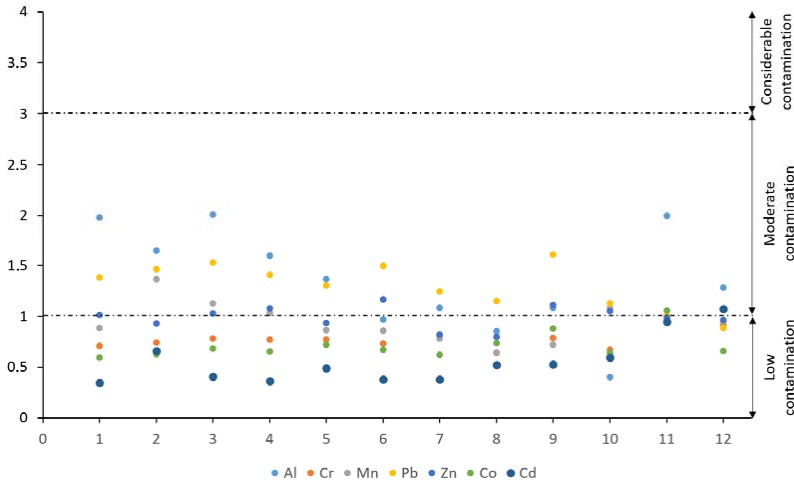

**Figure 2.** Sediment enrichment factor plot. Sampling sites labelled from 1 to 12 represent Stations 1 to 10 and core samples 1 and 3, respectively.

The *CF* results ranked the heavy metals from highest to lowest as follows: Al > Pb > Co > Zn > Cr > Mn > Cd. More specifically, the *CF* results indicate considerable Al, Pb, and Co contamination *CF* and moderate contamination for the remaining elements (Figure 3).

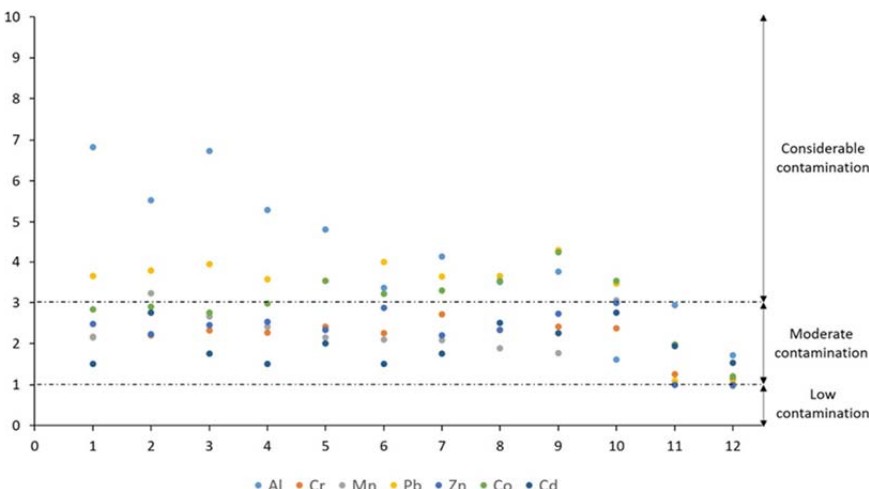

**Figure 3.** Sediment contamination factor plot. Sampling sites are labelled from 1 to 12 to represent Stations 1 to 10 and core samples 1 and 3, respectively.

The PLI results from the present study indicated a significant pollution hazard in the lake (Figure 4).

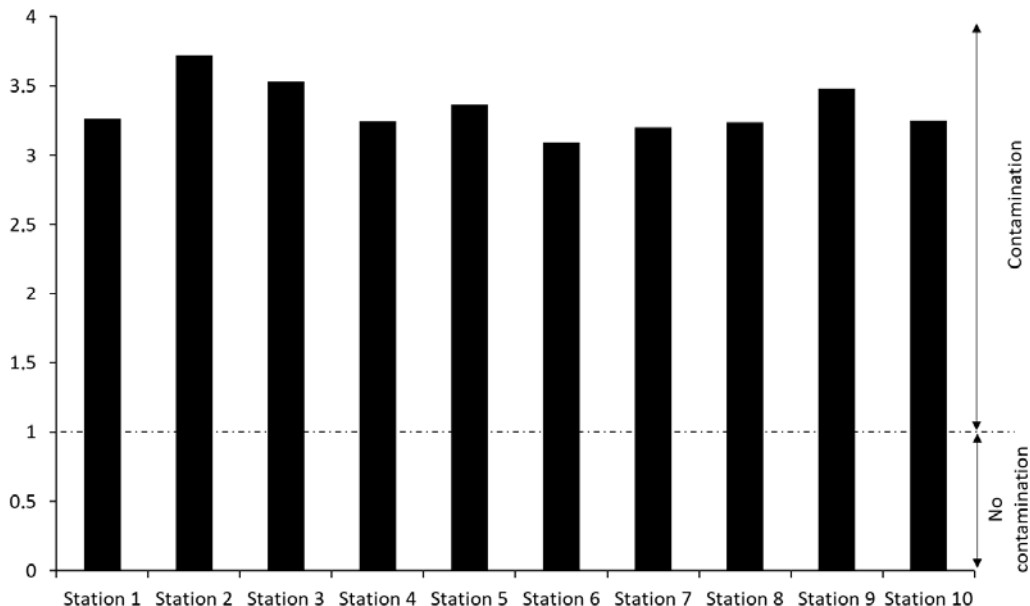

**Figure 4.** PLI values for Bafa Lake sampling stations.

### 3.4. Results of the Potential Ecological Risk Assessment (PERI)

Of the heavy metals examined in this study, the concentrations from the sediment samples for Cr and Ni were higher than both the TEC and PEC values from the SQGs (Table 7).

The indices were also used to assess the potential harm from heavy metal contamination (Table 8). Considering $E_r^i$, Cr, Mn, Pb, and Zn were categorized as low risk, while Cd was moderate risk. With regard to PERI, Stations 2, 8, 9, and 10 were categorized as moderate risk.

**Table 7.** Heavy metal concentrations in sediment (mg$^{-1}$ kg) and SQG [a] values.

|  | **Al** | **Cr** | **Mn** | **Pb** | **Zn** | **Co** | **Cd** | **Ni** | **Fe** |
|---|---|---|---|---|---|---|---|---|---|
| Station 1 | 5621.17 | 209.79 | 647.68 | 14.37 | 55.50 | 0.36 | 0.06 | 239.11 | 31,232.00 |
| Station 2 | 4553.50 | 214.63 | 973.15 | 14.87 | 49.76 | 0.37 | 0.11 | 210.57 | 30,500.39 |
| Station 3 | 5547.04 | 225.66 | 802.74 | 15.49 | 54.80 | 0.40 | 0.07 | 222.94 | 30,435.12 |
| Station 4 | 4352.73 | 219.90 | 725.87 | 14.05 | 56.66 | 0.38 | 0.06 | 229.83 | 29,984.75 |
| Station 5 | 3961.52 | 234.70 | 645.45 | 13.91 | 52.31 | 0.45 | 0.08 | 173.02 | 32,023.06 |
| Station 6 | 2765.01 | 218.84 | 631.77 | 15.70 | 64.43 | 0.41 | 0.06 | 120.41 | 31,540.79 |
| Station 7 | 3396.12 | 263.87 | 627.61 | 14.30 | 49.35 | 0.42 | 0.07 | 114.02 | 34,457.03 |
| Station 8 | 2888.18 | 227.39 | 564.62 | 14.34 | 52.29 | 0.54 | 0.10 | 112.15 | 37,448.14 |
| Station 9 | 3091.53 | 235.09 | 531.41 | 16.84 | 61.18 | 0.54 | 0.09 | 36.66 | 31,489.62 |
| Station 10 | 1324.97 | 231.25 | 919.62 | 13.61 | 66.90 | 0.45 | 0.11 | 52.56 | 36,264.39 |
| C1 | 7252.70 | 379.44 | 687.55 | 12.79 | 68.73 | 0.82 | 0.20 | - | 39,930.00 |
| C2 | 3478.75 | 364.34 | 710.32 | 12.80 | 67.16 | 0.75 | 0.20 | - | 38,300.00 |
| C3 | 4765.98 | 357.60 | 800.33 | 12.80 | 69.36 | 0.53 | 0.23 | - | 41,023.00 |
| Avg. | 4076.86 | 260.19 | 712.93 | 14.30 | 59.11 | 0.49 | 0.11 | 151.13 | 34,202.18 |
| TEC * | - | 43.40 | - | 35.8 | 121.00 | - | 0.99 | 22.70 | - |
| PEC * | - | 111.00 | - | 128.00 | 459.00 | - | 4.99 | 48.60 | - |

TEC *: threshold effect concentration. PEC *: probable effect concentration. [a] MacDonald et al. (2000) [59].

**Table 8.** Ecological risk levels for each heavy metal ($E_r^i$) and potential ecological risk assessment (PERI) of sediment samples by sampling station.

|  | **Cr** | **Mn** | **Pb** | **Zn** | **Cd** | **PERI** | **Grade of PERI** |
|---|---|---|---|---|---|---|---|
| Station 1 | 4.31 | 2.15 | 18.28 | 2.48 | 45.00 | 72.22 | low risk |
| Station 2 | 4.41 | 3.23 | 18.92 | 2.22 | 82.50 | 111.28 | moderate risk |
| Station 3 | 4.63 | 2.67 | 19.71 | 2.45 | 52.50 | 81.96 | low risk |
| Station 4 | 4.52 | 2.41 | 17.88 | 2.53 | 45.00 | 72.33 | low risk |
| Station 5 | 4.82 | 2.14 | 17.70 | 2.34 | 60.00 | 87.00 | low risk |
| Station 6 | 4.49 | 2.10 | 19.97 | 2.88 | 45.00 | 74.45 | low risk |
| Station 7 | 5.42 | 2.09 | 18.19 | 2.20 | 52.50 | 80.40 | low risk |
| Station 8 | 4.67 | 1.88 | 18.24 | 2.34 | 75.00 | 102.13 | moderate risk |
| Station 9 | 4.83 | 1.77 | 21.42 | 2.73 | 67.50 | 98.25 | moderate risk |
| Station 10 | 4.75 | 3.06 | 17.32 | 2.99 | 82.50 | 110.61 | moderate risk |
| C1 | 2.50 | 1.07 | 5.43 | 1.00 | 58.00 | 68.00 | low risk |
| C3 | 2.28 | 0.96 | 5.21 | 1.00 | 45.75 | 55.21 | low risk |
| $E_r^i$ Avg. | 4.30 | 2.13 | 16.52 | 2.26 | 59.27 |  |  |
| Grade of $E_r^i$ | low risk | low risk | low risk | low risk | moderate risk |  |  |

### 3.5. PCA

PCA was performed for all the environmental and water variables and the heavy metal sediment concentrations. The total variance was 75.392% (Figure 5). All the data used for this analysis is included in the supplementary files (see Supplementary File). The PCA showed that, in the first component, the most influential variables in the lake ecosystem were pH, temperature, and DO, while in the second component, they were Co, conductivity, Cd, TDS, and DO.

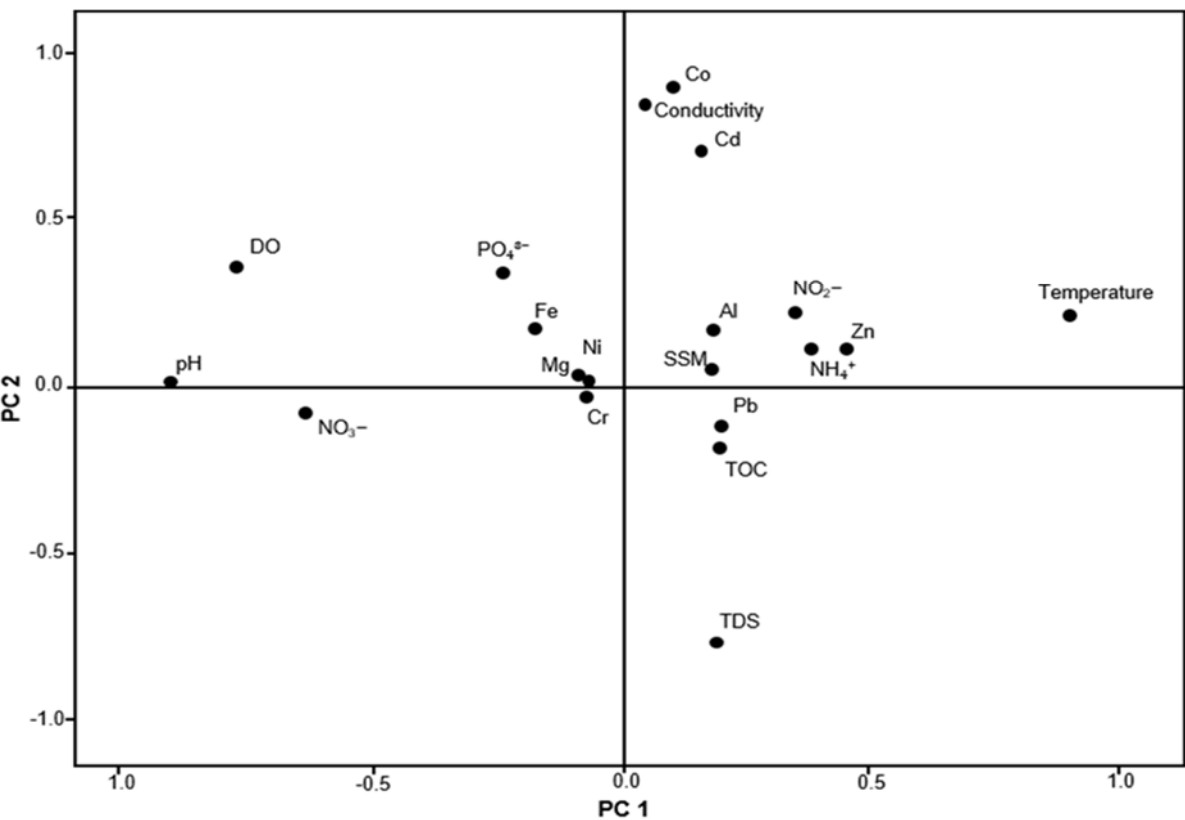

**Figure 5.** PCA analysis results.

## 4. Discussion

This study reported the accumulation levels of nine heavy metals (Al, Cd, Co, Cr, Fe, Mn, Ni, Pb, and Zn) to assess the quality of the sediment in Lake Bafa, one of the most important wetlands in Turkey's Aegean region. The lake's sediment quality was examined in terms of TEC and PEC values, PERI, and $NI_{geo}$, *EF*, *CF*, and PLI indices. Lake Bafa was selected as the sampling area because of its unique aquatic ecosystem being connected to both the sea and the groundwater system. In addition, the lake is affected by several anthropogenic factors, particularly agricultural, domestic, and industrial waste. The most important problems facing the Büyük Menderes basin are the salinization of the land due to the rapidly developing industry, high rates of fertilizer use in agricultural areas, and excessive use of groundwater for irrigation [64]. Recent studies of the Büyük Menderes water quality define it as polluted [65–68].

The mean heavy metal concentrations in the surface sediment of Lake Bafa ranked as follows from highest to lowest: Fe > Al > Mn > Ni > Cr > Zn > Pb > Co > Cd. These results are consistent with those from previous studies of the same area [69,70] and other studies conducted in Turkey (Table 9). Considering each element separately, for example, Al sediment concentrations in the area where Büyük Menderes River enters Lake Bafa (Station 5) differed significantly from other stations. Similarly, previous research reported Al pollution in the Büyük Menderes River water column [68]. This suggests that Al pollution of Lake Bafa may be coming from the Büyük Menderes River. In this regard, it was thought that Lake Bafa might be polluted by the Büyük Menderes River in terms of the Al element. The statistical analyses indicated that Mn accumulated especially near olive oil factories (Station 9), while Ni values were highest in the surface sediment near olive oil factories (Station 9) and the hotel (Station 10). Previous studies showed that the liquid and solid wastes released during olive oil processing contained high Mn and Ni levels [71–75]. This suggested that the high Mn and Ni concentrations detected especially in these areas of Lake Bafa were caused by waste contaminated with these two elements

from olive oil factories. Lake Bafa's water budget is not known exactly and there is no definite information in the literature about underground water resources entering the lake from the lake floor. A previous study examined the groundwater quality parameters and heavy metal concentrations in the region of the Büyük Menderes River between Lake Bafa and the Aegean Sea coast [64]. The high Zn concentrations (134 mg L$^{-1}$) were reported in groundwater taken from the station close to Lake Bafa. Thus, seasonal changes in Zn concentration within the lake may be due to groundwater.

**Table 9.** Heavy metal values measured in sediment in similar studies carried out in Turkey (mg kg$^{-1}$ dry weight).

| | Al | Cd | Co | Cr | Cu | References |
|---|---|---|---|---|---|---|
| Lake İznik (Bursa) | 5.6 (%) | 0.20 | 13.50 | 63.70 | 25.40 | [76] |
| Lake Yeniçağ | - | 0.8 | - | 92.8–274.2 | - | [77] |
| Lake Karataş (Burdur) | - | 0.20 | - | 37.56 | 21.95 | [78] |
| Lake Işıklı | - | 0.09–0.30 | - | - | 5.12–16.48 | [79] |
| Lake Beyşehir (Konya) | - | - | - | - | 7.16 | [80] |
| Lake Kovada (Isparta) | 6672.50 | 0.11 | - | 11.59 | 7.82 | [81] |
| Lake Bafa | - | - | - | 181–388 | 19.48–62.18 | [69] |
| Lake Bafa | - | 0.18 | - | 259.20 | - | [27] |
| Lake Bafa * | - | - | 18.35 | 88.95 | 35.50 | [70] |
| Lake Bafa | 54.03–7251.38 | 0.02–0.20 | LOD-0.73 | LOD-284.50 | - | This study |
| | **Fe** | **Mn** | **Ni** | **Pb** | **Zn** | **References** |
| Lake İznik | 3.0 (%) | 791.40 | 38.30 | 17.60 | 68.30 | [76] |
| Lake Yeniçağ | - | 756–1143 | - | 8.4–16 | - | [77] |
| Lake Karataş (Burdur) | 6968.62 | 306.89 | 131.81 | 0.88 | 28.16 | [78] |
| Lake Işıklı | 1721.26–7700.27 | 135.98–556.95 | 11.55–38.93 | 0.91–5.50 | 12.01–28.87 | [79] |
| Lake Beyşehir (Konya) | 10,390.59 | 484.19 | - | - | 39.82 | [80] |
| Lake Kovada (Isparta) | 5030 | 107.85 | 15.87 | 2.93 | 21.83 | [81] |
| Lake Bafa | 2.62–3.91 (%) | 625–1181 | 153–514 | 6.09–35.5 | 56.02–116.14 | [69] |
| Lake Bafa | 36,266.95 | 703.08 | 307.80 | 17.28 | 78.84 | [27] |
| Lake Bafa * | 32,850 | 557.25 | 195.00 | - | 42.05 | [70] |
| Lake Bafa | 2.23–4.13 (%) | 197.05–1331.47 | 115.96–373.48 | 11.02–27.57 | 22.03–87.17 | This study |

* Highest seasonal mean concentrations.

As outlined earlier, the lake used to be a bay in the Aegean Sea before the expansion of the Büyük Menderes delta turned it into a lake, which implies that Lake Bafa can be regarded as a sediment trap. There is a continuous transport of sediment from the Büyük Menderes River to Lake Bafa. Considering that the heaviest rainfall in the area occurs during the fall and spring seasons, the hydrodynamic conditions of the Büyük Menderes River increase, making sediment transport more rapid, thereby causing more pollution due to heavy metal inflows. Of the various estimates made for Lake Bafa's sedimentation rate [28,41,69], the mean value is 0.36 cm y$^{-1}$. In other words, the core sample taken for the present study represents the last 30 years of accumulated sediment in the lake. The analysis showed that the core samples from 5–10 cm depth adsorbed more metal than those taken between 0–5 cm depth. However, the difference was not significant in terms of statistical analyses. It seemed that the pollution load pattern had been regular for a certain time.

TOC levels in Lake Bafa sediment varied between 0.59 and 5.50 g kg$^{-1}$. In a previous study, similar mean TOC values (3.2 g kg$^{-1}$) sampled from two different points were reported [82]. As a result of the correlation analyses between TOC and heavy metal

concentrations in the Lake Bafa sediment, a positive correlation was determined between the amount of Cr and the amount of TOC. A strong positive correlation was also found between Al and Ni, and between Cd and Co. In a study of Lake Iznik, strong correlations were also reported between Al-Ni and Cd-Co levels in sediment samples [76].

Considering $NI_{geo}$ for Ni and Mn, some stations were slightly polluted, while most stations were very slightly polluted for Cr and Co. For Cd, half of the stations were slightly polluted, while the other half were very slightly polluted. It was observed that in terms of Al, two stations where the Büyük Menderes River enters Bafa Lake and $C_1$ core samples were very slightly polluted. Since there was a strong positive correlation between Al-Ni concentrations, it can be interpreted that Al might have a similar pollution potential in all stations in the future where Ni accumulates. In a similar study of Lake Bafa using $NI_{geo}$, almost all the surface and core sediments were categorized as unpolluted for Fe, Cr, Mn, Pb, Ni, Zn, and Cu, and lightly polluted for Hg [69]. In another previous study, it was found that some sediment samples taken from shallow areas of Lake Bafa were contaminated with Cd and Ni [70].

For all stations, the *EF* values indicated no anthropogenic enrichment (*EF* < 2) of heavy metals at most sites. The heavy metals with the highest *EF* values were Al and Pb, indicating that Al concentrations are affected by anthropogenic factors, while Pb concentrations are on the verge of being so. That is, the lake is being polluted with these two heavy metals. In addition, the correlation matrix indicates the lake's sediment is at risk of Ni pollution in the coming years due to the strong positive relationship between Al and Ni levels. Consistent with previous studies for this area, the *EF* values did not indicate pollution in the lake sediment for the other tested metals. In parallel to the $NI_{geo}$ and *EF* results, *CF* results indicate considerable Al, Pb, and Co contamination *CF* and moderate contamination for the remaining elements. That is, the lake is facing heavy metal pollution. A strong Cd contamination and moderate Ni contamination in certain locations of Lake Bafa were previously reported [70], while another study reported low Pb, Cu, Zn, and Mn contamination, and moderate Hg contamination at some stations [69].

The PLI results from the present study indicated a significant pollution hazard in the lake. The surface sediment was polluted by heavy metals, probably from anthropogenic sources. This sedimentary contamination may be caused by various factors: intense industrial activity in the Büyük Menderes region, domestic sewage outflow from settlements with no sewage infrastructure, waste from olive oil factories and tourist facilities, phosphate fertilizer and pesticide run off from agricultural areas, and heavy traffic on the Milas-Soke highway. Similar PLI findings from sampling points in Lake Bafa were also revealed in a previous study [70].

Of the heavy metals examined in this study, the concentrations from the sediment samples for Cr and Ni were higher than both the TEC and PEC values from the SQG's. This indicates that Cr and Ni are likely to be having harmful effects on aquatic organisms in Lake Bafa. Although Ni is a naturally occurring element in the strata of the Büyük Menderes delta, where Lake Bafa is located [69,83], the excessive Ni concentrations cannot be of natural origin alone (both natural and anthropogenic sources). Although it is not bioaccumulated in natural ecosystems [84], the high surface sediment Ni concentrations may have anthropogenic sources, such as mining and mineral processing waste, emissions from fossil fuel vehicles, domestic and industrial waste, and organic and inorganic agricultural outputs [70,84,85]. Previous studies have also reported high Cr concentrations in Lake Bafa sediment [28,69,86]. The most important known pollutant sources of Cr are untreated domestic and industrial wastes [87], particularly leather industry wastewater originating from Uşak and Aydın Karacasu, which enters the lake via the Büyük Menderes River. The PERI results indicated that sampling stations in the east-southeast part of the lake, where the residential areas, tourism activities (hotel and restaurant area), and olive oil factories were located, and Station 2, which is the closest station to the Muğla Aydın highway, carried moderate risk. According to the PERI results, it can be said that the accumulation trend in the lake was in the southwest line. The PCA showed that, in the first component, the

most influential variables in the lake ecosystem were pH, temperature, and DO, while in the second component, they were Co, conductivity, Cd, TDS, and DO.

## 5. Conclusions

The present study identifies the spatio-temporality of HMs in the sediments of Lake Bafa, Turkey. This study is one of the most comprehensive investigations using *NI$_{geo}$*, *EF*, *CF*, PLI, and PERI for Lake Bafa, which is one of the most important wetlands of Turkey's Aegean region. By comparing the obtained findings with the results of the studies that have been done and will be done in wetlands with similar characteristics both in Turkey and in the world, meaningful scientific inferences can be made about the environmental fates of such toxic pollutants. Results revealed very significant results regarding the ecological sustainability of Lake Bafa. It was observed that the lake sediment has been under the pressure of heavy metal pollution. According to the results of the risk assessments, the concentrations of Al, Ni, Cr, Co, and Cd in the lake sediment may reach levels that will endanger the ecosystem in the future, and this accumulation is especially concentrated in the southwestern part of the lake. In addition, both in this study and in other similar previous studies, the accumulation of Al in the sediment was highlighted as very important, and it was emphasized that the most important source of this pollution was the Büyük Menderes River. For this reason, it is significant that both the local authorities take decisions in the short term, and the studies to be carried out in the long term should focus on preventing the accumulation of these heavy metals and their possible sources. To ensure the future sustainability of the lake's ecosystem, it is important that such studies are conducted periodically to determine the lake's ecological status, monitor changes in pollutant levels, and thereby provide early warning of future problems. In terms of practical implications, local and national authorities should always be aware of the lake's ecological status and spare financial resources for scientific research in the area.

**Supplementary Materials:** The following supporting information can be downloaded at: https://www.mdpi.com/article/10.3390/su15139969/s1.

**Author Contributions:** Conceptualization, A.Y. and M.Y.; methodology, A.Y. and M.Y. software, A.Y.; validation, A.Y. and M.Y.; formal analysis, A.Y.; investigation, A.Y.; resources, A.Y.; data curation, A.Y.; writing—original draft preparation, A.Y.; writing—review and editing, A.Y.; visualization, A.Y.; supervision, M.Y. All authors have read and agreed to the published version of the manuscript.

**Funding:** This research was supported by Muğla Sıtkı Koçman University Coordinatorship of Scientific Research (BAP 13/121) and Muğla Sıtkı Koçman University Coordinatorship of Teaching Staff Training Program (OYP). Also, it is based on the Ph.D. thesis of the corresponding author.

**Data Availability Statement:** Data for this study are available upon request to the corresponding author.

**Conflicts of Interest:** The authors declare no conflict of interest.

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
