# Peer review of "Heavy Metal Contamination and Potential Ecological Risk Assessment in Sediments of Lake Bafa (Turkey)"

_sustainability, doi:10.3390/su15139969_

Round 1

Reviewer 1 Report

This topic is important to assess the quality of aquatic environment, and the research content and results are worthy of affirmation. However, there are still some problems in the manuscript.

·       Supplementary table 1: average (monthly or seasonally). The same comments also were noticed for Supplementary tables 3& 5& 7& 9& 11& 12& 14& 16.

·       “Which can be toxic to organisms at high concentrations [14]” please, can you read the cited reference again? I can send it to you if you want

·       What is the purpose of using so many similar evaluation metrics?

·       From obtained data, some sites contain concentrations of certain metals up to 41g/kg, is it true?

·       SSM in material & Method while TDS in the results

·       Why sediment samples were homogenized in porcelain mortar before sieving?

·       Need explanation, how in table 7 the concentrations of heavy metal were very higher in core samples compared with other samples, while in table 6 , NIgeo of core samples were < 1, that is mean unpolluted ?

·       Some results are not well discussed!

Dear Author

Some sentences are not clear to readers!

Reviewer 2 Report

The manuscript “Heavy metal contamination and potential ecological risk as-sessment in sediments of Lake Bafa (Turkey)” presents spatial and temporal variations of nine heavy metal concentrations in the sediments of Lake Bafa in Turkey’s wetland region. The study also provided a potential ecological risk assessment (PERI) based on geoaccumulation index (NIgeo), enrichment factor (EF), contamination factor (CF), and pollution load index (PLI). The results are significant for the selected region in order to determine the lake’s ecological status, as well as changes in pollutant levels, and to assess the future problems in sediments pollution. The study is of local interest; therefore the authors need to emphasize the importance of the study on the global level in the last paragraph of introduction as well as in conclusions. Also, in conclusions, authors need to add the possibility and focus for future studies. The results are clearly presented and discussed. The manuscript needs minor revision.

Reviewer 3 Report

In the manuscript titled “Heavy metal contamination and potential ecological risk assessment in sediments of Lake Bafa (Turkey)” the authors examined the spatio-temporality of heavy metal concentrations (Al, Cd, Co, Cr, Fe, Mn, Ni, Pb and Zn) in the sediments of Lake Bafa, one of the most important wetlands of Turkey’s Aegean region. This paper is interesting as heavy metals are persistent and ubiquitous pollutants which in high concentrations can cause serious damage to human health and aquatic ecosystems. Despite all this paper presents some problems that should be solved before the definitive publication. My opinion is to accept the paper with major revisions.

Major points:

1)     Title: The title gives too much regional impact to the manuscript. I suggest the authors to edit it.

2)     An important limitation for this study is that the authors considered only the sediment matrix to evaluate heavy metal concentrations. For a complete assessment of the state of the aquatic ecosystem studied, the authors should also consider the water matrix (Al, Cd, Co, Cr, Fe, Mn, Ni, Pb, and Zn).

3)     Conclusions: I suggest the authors extend this paragraph and report the salient points of the paper done.

4)     References: Authors should adapt the bibliography according to the journal’s guidelines for authors.

Round 2

Reviewer 3 Report

The manuscript has been improved and all suggestions of the reviewers have been carried out. My opinion is to accept the paper in current form for publication.